# LQFM289: Electrochemical and Computational Studies of a New Trimetozine Analogue for Anxiety Treatment

**DOI:** 10.3390/ijms241914575

**Published:** 2023-09-26

**Authors:** Jhon K. A. Pereira, André G. C. Costa, Edson S. B. Rodrigues, Isaac Y. L. Macêdo, Marx O. A. Pereira, Ricardo Menegatti, Severino C. B. de Oliveira, Freddy Guimarães, Luciano M. Lião, José R. Sabino, Eric de S. Gil

**Affiliations:** 1Faculty of Pharmacy, Federal University of Goias, Goiânia 74690-970, Brazil; jhonkennedy@discente.ufg.br (J.K.A.P.); andrexgabriel@egresso.ufg.br (A.G.C.C.); edson.silvio.b@gmail.com (E.S.B.R.); isaacyvesl@gmail.com (I.Y.L.M.); omarx@discente.ufg.br (M.O.A.P.); rm_rj@ufg.br (R.M.); 2Departament of Chemistry, Federal Rural University of Pernambuco, Recife 52171-900, Brazil; severino.oliveira@ufrpe.br; 3Institute of Chemistry, Federal University of Goias, Goiânia 74690-970, Brazil; freddy@ufg.br (F.G.); lucianoliao@ufg.br (L.M.L.); 4Institute of Physics, Federal University of Goias, Goiânia 74690-970, Brazil; jrsabino@gmail.com

**Keywords:** drug discovery, anxiety, antioxidant, oxidation mechanism, butylhydroxytoluene

## Abstract

This study employs electrochemical and Density Functional Theory (DFT) calculation approaches to investigate the potential of a novel analogue of trimetozine (TMZ) antioxidant profile. The correlation between oxidative stress and psychological disorders indicates that antioxidants may be an effective alternative treatment option. Butylatedhydroxytoluene (BHT) is a synthetic antioxidant widely used in industry. The BHT-TMZ compound derived from molecular hybridization, known as LQFM289, has shown promising results in early trials, and this study aims to elucidate its electrochemical properties to further support its potential as a therapeutic agent. The electrochemical behavior of LQFM289 was investigated using voltammetry and a mechanism for the redox process was proposed based on the compound’s behavior. LQFM289 exhibits two distinct oxidation peaks: the first peak, *E*_p1a_ ≈ 0.49, corresponds to the oxidation of the phenolic fraction (BHT), and the second peak, *E*_p2a_ ≈ 1.2 V (vs. Ag/AgCl/KCl_sat_), denotes the oxidation of the amino group from morpholine. Electroanalysis was used to identify the redox potentials of the compound, providing insight into its reactivity and stability in different environments. A redox mechanism was proposed based on the resulting peak potentials. The DFT calculation elucidates the electronic structure of LQFM289, resembling the precursors of molecular hybridization (BHT and TMZ), which may also dictate the pharmacophoric performance.

## 1. Introduction

According to the World Health Organization (WHO), stressful or emotionally draining situations can lead to the emergence of mental disorders [1]. In 2015, about 3.6% and 4.4% of the world’s population suffered from anxiety disorders and depression, respectively [2,3].

Anxiety disorders are linked to fear and excessive restlessness [4,5]. Such emotional states decrease quality of life, causing other diseases with higher risk factors for individual and collective health, and thus resulting in socioeconomic impacts [5,6].

Although the exact causes of anxiety disorders remain unclear, growing clinical and preclinical evidence suggests that oxidative stress may play a crucial role in anxiety pathology [7,8]. An increase in reactive oxygen species (ROS), primarily O_2_^•−^ and H_2_O_2_, generated by both external and intrinsic factors, including the mitochondria, the excitotoxicity of glutamate, and the autoxidation of metabolites, is highly neurotoxic due to the vulnerability of brain cellular composition [8]. Currently, much evidence suggests that oxidative stress may be associated with many psychiatric disorders, including obsessive–compulsive disorder and panic disorder, both as a primary cause in some cases and as a consequence that may guide new treatments [7,8]. When there are high levels of ROS, it can compromise the production of dopamine, norepinephrine, serotonin and nitric oxide, through tetrahydrobiopterin oxidation. Tetrahydrobiopterin is an important coenzyme, which participates in the biosynthesis of many neurotransmitters. Accordingly, compounds with antioxidant profiles should help to reverse mood disorders [9]. In a recent paper, we showed the relationship between anxiety and an anti-inflammatory antioxidant compound [10].

Moreira et al. (2023) [10] showed that in a lipopolysaccharide (LPS)-induced neuroinflammation model, LQFM212 treatment reverted changes caused by LPS, demonstrated by attenuated anxiogenic- and depressive-like behaviors, reduced pro-inflammatory cytokines (TNF-α and IL-1β) and increased anti-inflammatory cytokines (IL-4 and IL-10) on serum, and also improved oxidative stress-related changes (levels of nitrite, malondialdehyde, glutathione and carbonylated protein, and superoxide dismutase, catalase, myeloperoxidase and cholinesterase activities) on the brain cortex and hippocampus. However, LQFM212 treatment did not attenuate the inflammatory changes in an LPS-induced pleurisy model [10]. LQFM212 and LQFM289 (**3**) have a 2,6-di-tert-butylphenol scaffold, which is responsible for their antioxidant profiles.

Growth differentiation factor 11 (GDF11), which acts as an anti-inflammatory cytokine, has recently been shown to be a promising target for new prototype drugs, as it is related to the ability to renew old and damaged components of neurons and a decline in its levels can be seen as an individual ages or suffers from neurological problems [11]. Similarly, the use of antioxidants is a potential alternative therapeutic option, either synergistically, with traditional anxiolytics, or individually through diet, supplementation, or medication [12,13]. Atmaca et al. (2004) [14] found that citalopram had antioxidant effects as an anxiolytic treatment in patients with social phobia. Some polyphenols have a partial agonist action that may produce anxiolytic-like effects without the side effects of total agonists like benzodiazepines [7,15,16].

Anxiolytic medications are commonly based on serotonin reuptake inhibitors or norepinephrine inhibitors; however, benzodiazepines may improve gamma-aminobutyric acid (GABA) binding in GABA receptors [17]. Despite current pharmacotherapy, about 30% of patients with psychological disorders remain unresponsive to treatment [7,11]. Consequently, it is imperative to develop new drugs to tackle this issue. In medicinal chemistry, molecular hybridization is a classic strategy to integrate structures of different bioactive compounds into a singular compound to maximize pharmacological properties [18].

Trimethozine (TMZ) (**2**) is a mild sedative medication that was employed during the 1960s to address various mood disorders, including depression and anxiety, and has been studied for its potential to treat chronic pain and insomnia [19,20].

The structure of TMZ (**2**) consists of trimethoxybenzene bonded to morpholine B by a carbonyl moiety. Morpholine is a privileged structure, which exhibits substantial therapeutic activity, especially in the treatment of neurodegenerative disorders and cognitive impairment [21,22]. In order to optimize the pharmacophoric groups of TMZ, a new lead compound named LQFM289 (**3**) can be obtained through a molecular hybridization strategy by replacing the first trimethoxybenzene with a bioactive compound possessing a stronger antioxidant profile, butylhydroxytoluene (BHT) [23] (**1**), as represented in Figure 1. In addition to its antioxidant properties, which make it a common ingredient in food [24] and cosmetic products [25], some studies have revealed that BHT can enhance the potential effects of antidepressant drugs as well as anxiolytics in neurodegenerative models [26,27,28].

In a previous work, LQFM289 (**3**) 10 mg/kg induced anxiolytic-like activity in mice without interfering with the animals’ motor activity [27]. The anxiolytic-like effects of LQFM289 (**3**) (10 mg/kg) were attenuated by flumazenil pretreatment, which suggests the participation of benzodiazepine binding sites. The docking of LQFM289 (**3**) showed strong interactions with benzodiazepine binding sites and matched well with receptor binding data. LQFM289 (**3**) also lowered corticosterone and tumor necrosis factor alpha levels in LQFM289 (**3**)-treated mice at a single oral dose of 10 mg/kg, which suggests that the anxiolytic-like effect observed in this compound can modulate other pathways [27].

In addition, owing to the electroactivity of BHT (**1**) and its derivatives, electroanalysis emerges as a promising tool to evaluate its redox properties, which may be useful in complementary chemical characterization, and also to estimate the oxidative stability and antioxidant properties of the compound, as well as offering an analytical target for further quantitative determinations [28,29,30,31].

The redox properties are also related to the molecular orbital parameters and to the conformation of the molecule [29]. To assess these parameters, computational chemistry calculations, such as density functional theory (DFT), are excellent tools [29]. Also, due to the direct correlation between antioxidant activity and therapeutic effect, the evaluation of these parameters can provide information on the chemical components that may be responsible for the biological effect, paving the way for the discovery of new drugs through prototypes such as LQFM289.

Therefore, the aim of this work is to present the electrochemical characterization of the LQFM289 (**3**) lead compound and the homologous reagents BHT (**1**) and TMZ (**2**) in aqueous media on glassy carbon electrode (GCE) according to their redox behavior and antioxidant activity. Furthermore, DFT calculations will be applied to elucidate the electronic structure of the TMZ and LQFM289 molecules, in order to support the electrochemical experimental results.

## 2. Results 

### 2.1. Electrochemical Characterization of LQFM289 (***3***) Lead Compound

The cyclic voltammograms of LQFM289 (**3**), a prototype drug of TMZ (**2**), in glassy carbon electrode ranging from −0.1 to 1.4 V in pH 7.0 0.1 mM PBS are shown in Figure 2A. Two anodic peaks, 1a and 2a, can be observed at *E*_p1a_ ≈ 0.55 V and *E*_p2a_ ≈ 1.05 V vs. Ag/AgCl/KCl_sat_. On the reverse scan, a cathodic peak, 1c, is seen at *E*_p1c_ ≈ 0.42 V, related to the anodic process, and 1a is also in evidence (Figure 2A, solid black lines). 

The redox behavior of all anodic peaks at different scan rates was investigated, and the voltammograms and linear plot *I*_pa_ vs. υ^1/2^ are presented in Figure 2C,D. 

In order to establish connections between LQFM289 (**3**) peak potentials and electroactive groups, the cyclic (Figure 2A) and DP (Figure 2B) voltammograms of its drug analogue TMZ (**2**), as well as its molecular moieties, BHT (**1**) and morpholine, were performed in similar conditions (Figure 2A,B). 

Figure 3 depicts the molecular electrostatic potentials (MEP) map and occupied molecular orbitals of the studied molecular species: TMZ (**2**), LQFM289 (**3**) and oxidized LQFM289 (**3**). The oxidized LQFM289 (**3**) represents the LQFM289 (**3**) molecular rearrangement after an oxidative process with the loss of a hydrogen atom. This molecule is a doublet with neutral charge; it is a free radical. The MEP regions depicted in red represent moiety, with negative character ranging for the blue regions, which are positively charged.

The effect of pH on the electro-oxidation of the new drug candidate, LQFM-289 (**3**), was evaluated from pH 3 to pH 10 (Figure 4).

### 2.2. Quantitative Applications

Figure 5 presents the calibration curves obtained for LQFM289 (**3**) and TMZ (**2**) in pH 8.0 0.1 mM PBS using the GCE. 

### 2.3. Structural Characterization of LQFM289 (***3***)

The X-ray analysis of compound (**3**) showed two independent molecules in the asymmetric unit of the monoclinic unit cell, whose atoms were labelled with suffixes A and B (Figure 1B and Figure 6).

## 3. Discussion

### 3.1. Structural Characterization of LQFM289 (***3***)

The difference between the conformers resides in the morpholine ring conformation, giving rise to the coexistence of different chair conformations (Figure 6). Molecule B shows a structural disorder in the morpholine ring, which lowered the quality of the X-ray diffraction data, as noted in Table 1. Because of this structural disorder, only the coordinates of the molecule A were used for DFT calculations.

In order to show the different chair conformations adopted by the morpholine group, we superposed the molecules using the atoms C1A to C7A as matching atoms to calculate the transformation matrix. The superposition of the molecules A and B (Figure 6A) revealed the different chair conformations of the morpholine fragment after being superposed by a proper matrix transformation, with an RMSD of 0.023 Å for the matched atoms. The DFT-optimized molecule was also superposed to molecule A (Figure 6B), showing that the latter was close to the relaxed conformation, with an RMSD of the matched atoms of 0.022 Å. In Figure 6C, the superposition of the molecules A and TMZ (**2**) (RMSD of 0.035 Å) evidences a preferred conformation of the morpholine in this kind of compound, adopting the chair conformation observed in fragment B. In Figure 6C, TMZ was transformed by an improper matrix. A search in the Cambridge Structural Database WebCSD (version 2023.1.0) returned one entry of trimetozine compound (**2**) with CCDC refcode TUJTAV [32], whereas the search for the compound (**3**) has returned no entries (More positions are shown in Appendix A).

Bond distances and angles were in good agreement among the compounds analyzed in this study (see Appendix A). Exceptions are the torsion angles relating to the amide moiety and the benzyl group. The dihedral angles of this region of the molecules are compared in Table 2.

### 3.2. Computational Data and Electrochemical Characterization

The results presented in Section 2.2 indicate that all anodic processes of LQFM289 are independent. Indeed, when the potential range was shortened from 0.3 to 0.75 V, the peak potentials related to the redox pair 1a/1c remained constant, thus showing no interdependence. Also, no change occurred in anodic process 2a when cycling from 0.8 to 1.4 V (Figure 2A, dashed black lines).

Despite the undeniable displacement of peak potential values, the results indicate that anodic processes 1a and 2a in LQFM289 (**3**) are correlated to phenolic and nitrogen moieties, respectively.

It is very well known that BHT (**1**) exhibits reversible electrochemical behavior at peak potentials lower than 0.4 V (neutral pH), which in fact explains its widespread use as an antioxidant [24,27,28,29,30,31]. The reversibility of the BHT (**1**) moiety of LQFM 289 (**3**) was confirmed with SWV (Figure 2B, inset). On the other hand, the redox pair 1a/1c observed for BHT (**1**) (Figure 2, red lines) and LQFM 289 (**3**) (Figure 2, black lines) showed a shift of c.a. 0.3 V. The higher peak potentials observed for BHT (**1**) moiety in LQFM 289 (**3**) can be explained by means of steric and electronic effects. Indeed, the morpholine moiety, present in TMZ (**2**) and LQFM 289 (**3**), exerts a withdrawing force over the electron density that in combination with the inherent steric effect may hamper the electron transfer and diffusion from bulk solution to the electrode surface. 

In turn, owing to the methylation of the phenolic BHT (**1**) moiety, the redox pair 1a/1c does not occur in TMZ (**2**), whereas the anodic peak, 1a‴, occurs at a higher peak potential when compared to the correspondent peaks 1a″ and 2a in LQFM 289 (**3**) (Figure 2, black lines) and morpholine (Figure 2B, green line), respectively. 

From Figure 3, it is possible to see that the most negative regions are located over the oxygen atoms in the molecule and that LQFM289 (**3**) has almost neutral characteristics over the molecular chain in comparison to TMZ (**2**). The highest occupied molecular orbital (HOMO) of all compounds has its localization over the aromatic ring of the BHT (**1**) compound, with energies around 7.34 EV. However, comparing HOMO-1 and HOMO-2 in the TMZ (**2**) and LQFM289 (**3**) compounds shows an inversion of the orbital localization in relation to their energies from the BHT (**1**) aromatic ring to the morpholine moiety. The HOMO-2 orbital of the TMZ (**2**) and HOMO-1 orbitals of the LQFM289 (**3**) counterparts are localized in the morpholine moiety, with energies ranging from 8.08 to 8.40 EV. The HOMO-1 orbital of the TMZ (**2**) and HOMO-2 orbitals of the LQFMs (**3**) are localized in the BHT (**1**) aromatic ring [28,29]. 

Hence, it can be inferred that the more homogeneous electron density distribution in LQFM289 (**3**) is responsible for the lower gap between anodic processes 1a and 2a (Figure 2A,B and Figure 3). Also, the first process is quasi-reversible, with an *I_a_*/*I_c_* ratio of 1.8 and ΔEp of 280 mV (Figure 2A, black lines). Meanwhile, the anodic peak, 2a, is irreversible.

An expressive decay of peak current levels was also observed after successive scans (Figure 2A, inset). This fact suggests the formation of insulation film and that no electroactive products from electrochemical oxidation were formed at the electrode surface [28,29,30].

The peak currents obtained at different scan rates exhibited a linear behavior for all anodic peaks. Thus, a linear graph *I*_p_ vs. v^1/2^ (r^2^ = 0.99) is depicted in Figure 2C,D, indicating that the charge transfer is controlled by diffusion [2,28,29,30,31]. 

The effect of pH on redox behavior was carefully evaluated (Figure 4). The plot of anodic peak potential, 1a, showed a slope value of approximately 59 mV·pH^−1^ and a linear decay (r = −0.97), suggesting a one to one electron–proton transference process, as established in the Nernstian processes [2,28,29,30,31].

Nevertheless, the anodic process 2a at higher peak potentials showed chaotic behavior, which can be attributed to the insulating film observed in the inset of Figure 2A that hampers electron transfer, and also to pH independence. In fact, the electropolymerizing reactions at the electrode surface were shown to be far stronger at acid pH (Figure 7).

### 3.3. Proposed Electrooxidation Mechanism

A plausible electrochemical oxidation mechanism for LQFM289 (**3**) was also proposed (Figure 7). The first peak (at lower potential of ≈0.4 V) was related to phenolic (BHT-like) oxidation, in line with previous literature reports [31]. As shown in scheme I, the oxidation involves one electron being transferred stepwise through a one-electron process, leading to oxidation.

The second peak for the new compound at ≈1.2 V depicted in scheme II was in agreement with the proposed electrooxidation mechanism in the literature; this is the electrooxidation of morpholine using a boron-doped diamond electrode. The irreversible process can be explained because the radical produced is very short-lived and therefore does not support reverse electron transfer [8,32,33,34,35,36]. The mechanism proposed for morpholine oxidation is illustrated in Figure 8 [32].

### 3.4. Quantitative Applications

Since the higher the pH, the lower the observed electrode fouling, pH 8.0 was chosen for the quantitative analysis of LQFM289 (**3**) and TMZ (**2**). The free electroactive phenolic moiety allows the exploration of peak 1a at a lower peak potential *E*_p1a_~0.45 V. However, for TMZ (**2**), the only peak, 1a′′′′, at a higher peak potential *E*_p1a_~1.25 V, is subject to a higher number of interfering species. 

Though the phenolic group confers higher sensibility, the unstable phenoxy intermediates undergo fast electropolymerization, even at low peak potentials. As a consequence, the insulating process leads to linearity compromise [28,29,30,31]. Hence, the limit of detection obtained for LQFM289 (**3**) and TMZ (**2**) was 10 µM and 10 mM, respectively, whereas the coefficient of linear correlation was 0.95 and 0.99, respectively.

## 4. Materials and Methods

### 4.1. Reagents, Samples and Solutions

Analytical grade butylhydroxytoluene and morpholine were purchased from Synth. Trimetozine (**2**) and the new lead compound, LQFM289 (**3**), were formed by the research group at the Laboratory of Medicinal Pharmaceutical Chemistry (LQFM) Faculty of Pharmacy, Federal University of Goiás/Goiânia/Brazil [27]. Solutions of 1 mmol·L^−1^ were prepared as absolute solvents prepared with DMSO at 4 °C. Acetate (A) and Phosphate (P) Buffer Solutions (BS) were used as electrolytes and prepared at different pH values, 3.0, 5.0, 7.0, 9.0, and 10.0, in 0.1 mM solutions with analytical purity reagents and deionized water.

### 4.2. Electrochemical Assays

Electrochemical analyses were carried out in a potentiostat/galvanostat PGSTAT^®^ (Metrohm Autolab, Utrecht, The Netherlands) integrated with NOVA 2.1^®^ software. The electrochemical experiments were carried out in a one-compartment glass electrochemical cell (1 mL) with a conventional three electrode system: a Ag/AgCl/KClsat (3 M KCl) electrode as the reference electrode, a platinum electrode as the auxiliary electrode, and a glassy carbon electrode (GCE), 3.0 mm diameter, as the working electrode. All experiments were carried out at ambient temperature (23 ± 2 °C) in triplicates (*n* = 3). All data were then analyzed and treated with Origin 8^®^ software.

Cyclic Voltammetry (CV) experiments were carried out with a glassy carbon electrode (GCE), 3.0 mm diameter, as the working electrode in a potential range of 0 to 1.4 V vs. Ag/AgCl/KClsat. The scanning rate was equal to 100 mV s^−1^. Prior to the measurement, the GCE was mechanically polished using an Al_2_O_3_ suspension, washed and cycled in a 1 mM NaOH solution until a stable voltammogram was obtained (approximately 8–10 cycles). For the mass transference study, experiments were carried out with different scanning rates of 25, 50, 100, 250 and 500 mV s^−1^; however, the potential window remained the same. 

Differential Pulse Voltammetry (DPV) was also performed using the GCE and the parameters applied were a potential range 0 to 1.3 V; a scan rate of 10 mV s^−1^; a pulse amplitude of 50 mV; and a pulse width of 70 ms. 

The following experimental conditions were applied for Square Wave Voltammetry (SWV): a pulse amplitude of 50 mV; a frequency of 50 Hz; and a potential increment of 2 mV (corresponding to a scan rate of 100 mV s^−1^).

### 4.3. Computational Studies

The electronic structure calculations were performed using Gaussian software 16.0 [37]. The calculations were carried out using the Density Functional Theory (DFT) method for all molecular systems, within the exchange correlation function M062-x [38] and the Def2-TZVP [39,40] basis set. The full geometry optimization was computed followed by normal mode frequency calculation to ensure that the minimal energy was found and that only real positive numbers were reported for all vibrational frequencies of all studied species. In addition, the Molecular Electrostatic Potentials (MEP) were computed using Molden software [41] https://www3.cmbi.umcn.nl/molden/ (accessed on 1 July 2023) and visualized with the Jmol program [42] http://jmol.sourceforge.net/ (accessed on 7 October 2016). The MEP gives the charge distribution around the molecule and is used to analyze and interpret the reactivity of molecular systems [43].

### 4.4. Crystal Structure Determinations

The single crystal X-ray structures of the compounds (**2**) and (**3**) were determined at room temperature. Suitable single crystals were obtained through slow evaporation using methanol. Data collections were performed using a Bruker-AXS Kappa Apex II Duo diffractometer operating with Mo-Kα (0.71073 Å) from an IµS micro-source, filtered by multi-layer mirror X-ray optics. Structure solution was obtained using Direct Methods implemented in Bruker software package SHELXTL [44] https://doi.org/10.1107/S2053273314026370 (accessed on 1 July 2023) and the final refinement was performed with full matrix least squares on F^2^ using SHELXL. The programs ORTEP-3 [45] and SHELXL [46] were used within the WinGX software package [47] https://doi.org/10.1107/S0021889899006020 (accessed on 1 July 2023). The refinement results, experimental details and crystal data are summarized in Table 1.

## 5. Conclusions

The results suggest that the LQFM289 (**3**) is electroactive in aqueous media on the GCE and that it undergoes anodic reactions in two stages, represented by peaks 1a and 2a. The first process is quasi-reversible and occurs with the removal of one electron and one proton from the phenolic group, from the BHT (**1**) portion, at Ep1a ~ 0.4 V. The second process is irreversible, at Ep1a ~ 1.1 V at pH 8.0, and occurs with the removal of one electron in the amino group from morpholine. The computational studies agreed with the electrochemical findings obtained for LQFM289 (**2**), its drug analogue TMZ (**2**), and its molecular counterparts, allowing the proposal of a redox mechanism. 

It is important to address that DFT calculations may not accurately represent the behavior of molecules in a complex system; however, the conjoint view of DFT paired with electroanalytical techniques yields valuable insights into oxidative behavior. Thus, despite the pharmacological effect not being considered in this study, it can be safe to assume that the displayed oxidative behavior of trimetozine, which was elucidated through electroanalytical and computational chemistry techniques, is related to its pharmacological behavior.

Therefore, these voltametric and DFT studies have demonstrated the pharmacological potential of LQFM289. However, different types of studies are still necessary, including traditional studies of toxicology in-vivo, involving the exposure of groups of animals, usually rats, to the compound. 

## Figures and Tables

**Figure 1 ijms-24-14575-f001:**
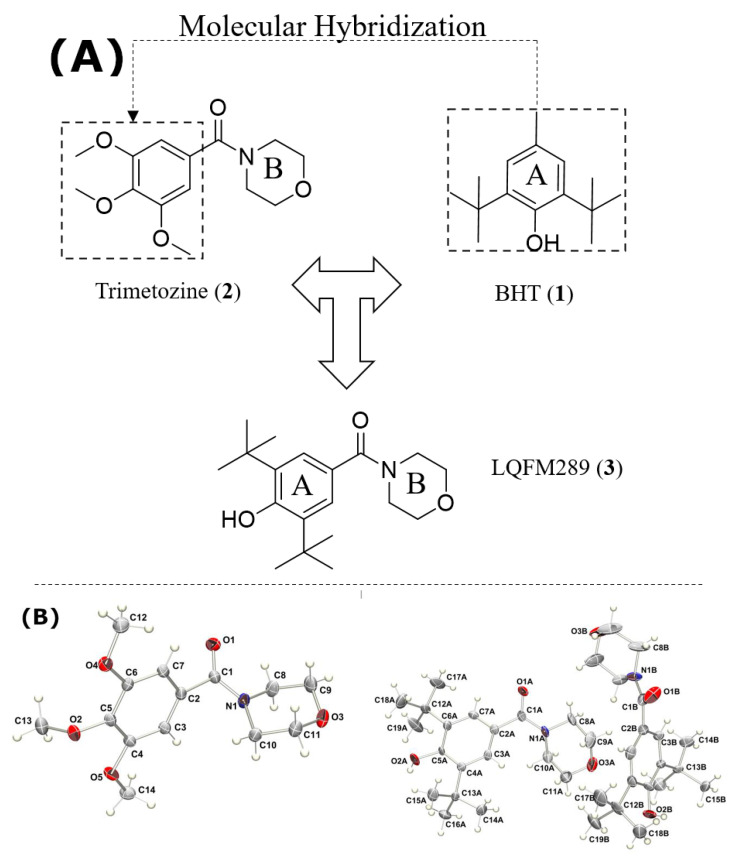
(**A**) Structural design of LQFM289 (**3**): BHT (**1**), trimetozine (TMZ) (**2**) lead compounds; (**B**) ORTEP view of the trimetozine (**2**) and LQFM289 (**3**).

**Figure 2 ijms-24-14575-f002:**
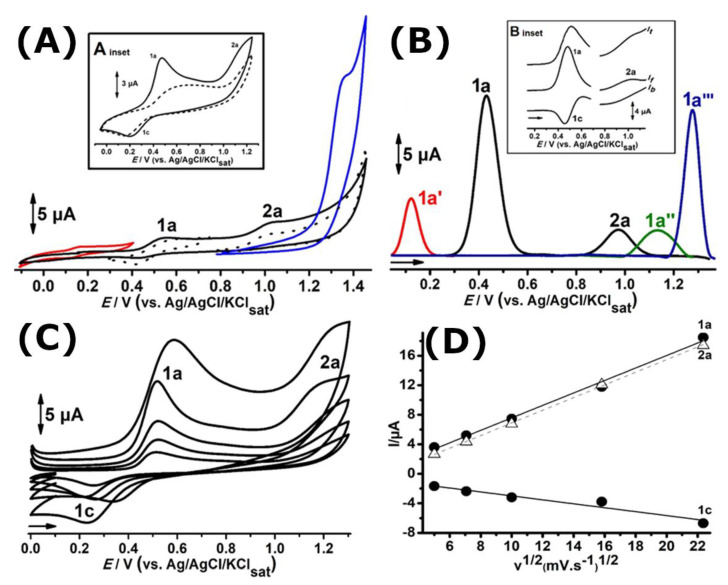
(**A**) Cyclic and (**B**) DP voltammograms for 0.1 mM solutions of LQFM 289 (**3**) (black lines), TMZ (**2**) (blue lines), BHT (**1**) (red lines) and morpholine (green line). (**C**) Cyclic voltammograms obtained for 0.1 mM LQFM289 (**3**) at different scan rates (25, 50, 100, 250, 500 mV·s^−1^). (**D**) Dependence of peak currents, 1a, 1c and 2a, on square root of scan rate obtained for 0.1 mM LQFM 289 in pH 7.0 0.1 M PBS. Inset: (**A**) Successive cyclic voltammograms obtained for 0.1 mM LQFM289 (**3**) 1st scan (–––) and 2nd scan (- - -); (**B**) SW voltammograms. All in pH 7.0 0.1 PBS at GCE.

**Figure 3 ijms-24-14575-f003:**
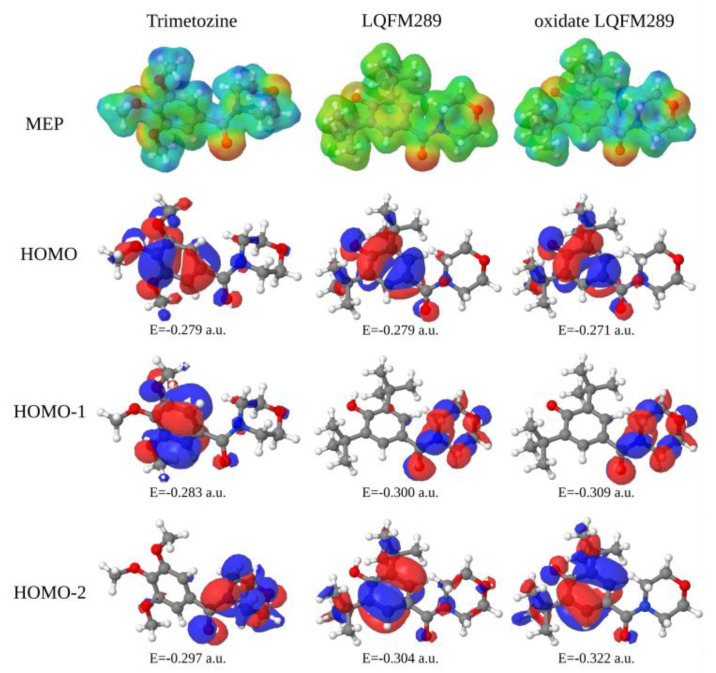
Molecular electrostatic potential maps and the tree lowest molecular orbital (HOMO, HOMO-1, and HOMO-2) energies of TMZ (**2**), LQFM289 (**3**), and oxidized LQFM289 (**3**). The MEPs were built with a cutoff of 0.05 and the molecular orbitals with a cutoff of 0.025.

**Figure 4 ijms-24-14575-f004:**
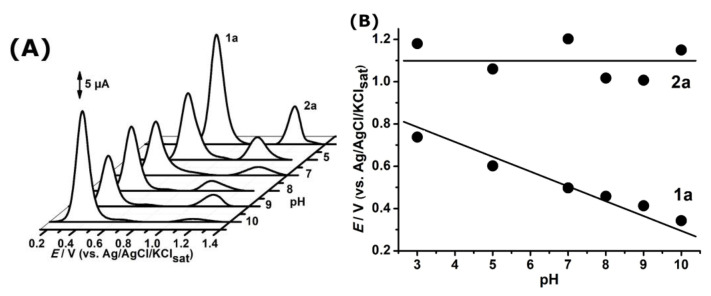
(**A**) Differential pulse voltammetry results of 0.1 mM LQFM289 (**3**) at pH range 3 to 10 in ABS and PBS. (**B**) Plots of *E*_pa_ × pH.

**Figure 5 ijms-24-14575-f005:**
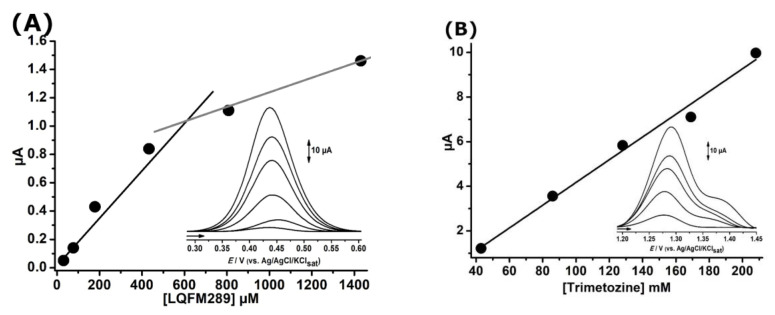
DP voltammograms and resulting calibration plot for increasing concentrations of LQFM 289 (**A**) and TMZ (**B**). All at GCE in pH 8.0 0.1 mM PBS.

**Figure 6 ijms-24-14575-f006:**
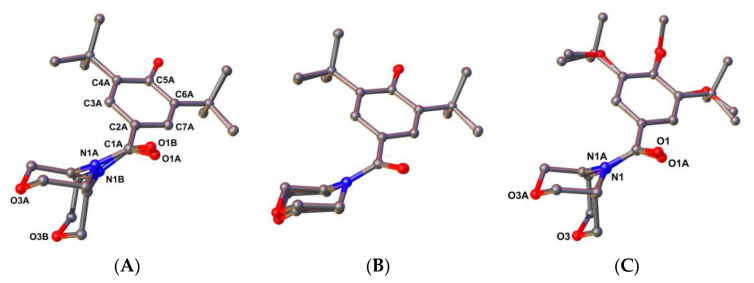
Molecule superposition using the matching atoms C1A to C7A: (**A**) independent molecules A and B of compound (**3**), (**B**) molecule A of compound (**3**) and the DFT optimized coordinates of the same molecule and (**C**) molecule A of compound (**3**) and the inversion-transformed TMZ (**2**) molecule.

**Figure 7 ijms-24-14575-f007:**
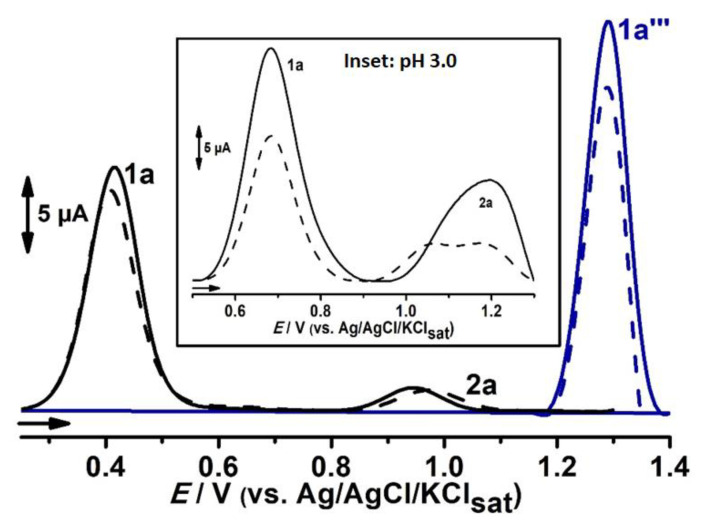
Successive DP voltammograms, 1st scan (solid line –––) and 2nd scan (dashed line - - -), obtained for 0.1 mM LQFM289 (**3**) (black line) in pH 9.0 and for 0.1 mM TMZ (**2**) (blue line) in pH 8.0. All in 0.1 M PBS at GCE. Inset: successive DP voltammograms obtained for 0.1 mM LQFM289 (**3**) in pH 3.

**Figure 8 ijms-24-14575-f008:**
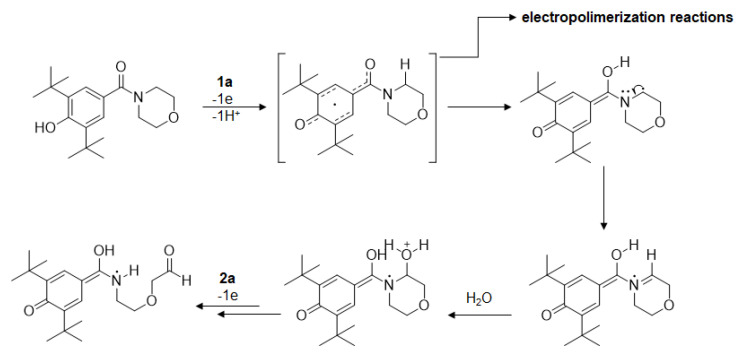
Proposed electro-oxidation mechanism: LQFM289 (**3**).

**Table 1 ijms-24-14575-t001:** X-ray data collection and refinement parameters for compounds (**2**) and (**3**).

Compound	(2)	(3)
Empirical formula	C14 H19 N O5	C19 H29 N O3
Formula weight	281.30	319.43
Crystal system	Triclinic	Monoclinic
Space group	P $\bar{1}$	P 2_1_/n
a (Å)	8.6525 (4)	17.3332 (14)
b (Å)	9.5424 (4)	12.2252 (9)
c (Å)	9.8544 (4)	18.9650 (15)
α (°)	64.2340 (10)	90
β (°)	77.059 (2)	110.427 (3)
γ (°)	71.725 (2)	90
V (Å^3^)	3692.12 (5)	3766.0 (5)
Z	2	8
ρ (Mg/m^3^)	1.350	1.127
µ (mm^−1^)	0.103	0.075
Absorption correction	multi-scan	multi-scan
Reflections collected	38,484	75,266
Unique reflections/R(int)	3918/0.0463	6655/0.1073
Completeness to θ = 25.0°	100.0%	100.0%
Data/restraints/parameters	3918/0/184	6655/0/430
Goodness-of-fit	1.130	1.084
Final R1/wR2 [I > 2 σ(I)]	0.0497/0.1466	0.1788/0.4337
R1/wR2 indices (all data)	0.0618/0.1529	0.2087/0.4521
Extinction coefficient	--	0.0041 (15)
Largest diff. peak and hole (e.Å^−3^)	0.359 and −0.226	0.880 and −0.449

**Table 2 ijms-24-14575-t002:** Selected torsion angles (°).

	LQFM289-A	LQFM289-B	TMZ	TUJTAV	289A-OPT
O1-C1-C2-C7	36 (1)	50 (1)	46.6 (2)	46.4 (2)	37.61
O1-C1-N1-C8	6 (1)	8 (1)	5.4 (2)	5.8 (2)	2.91
O1-C1-N1-C10	−154.0 (8)	−170.7 (9)	−158.1 (1)	−158.1 (1)	−149.03
C2-C1-N1-C10	30 (1)	13 (1)	27.3 (2)	26.8 (2)	34.15
C1-N1-C8-C10	−162 (1)	−179 (1)	−165.3 (2)	−165.7 (2)	−155.25
N1-C8-C9-O3	55 (1)	−56 (1)	−56.7 (2)	−55.7 (2)	55.8

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
