# Peer review of "LQFM289: Electrochemical and Computational Studies of a New Trimetozine Analogue for Anxiety Treatment"

_ijms, 2023, doi:10.3390/ijms241914575_

Round 1

Reviewer 1 Report

The studies described in the research article " LQFM289: Electrochemical and Computational Studies of a New Trimetozine Analogue for Anxiety Treatment" by Pereira et al have used electrochemical and Density Functional Theory (DFT) calculation approaches to test the potential of a novel analogue of trimetozine (TMZ) as a viable treatment option for anxiety disorders.

The following are some suggestions to improve the article:

1.       Line 57- Not all anxiolytic drugs target serotonin or norepinephrine. For example, Benzodiazepines facilitate GABA binding at GABA receptors.

2.       Line 64- Please change to “disorders”

3.       Line 185- Please correct “por”

4.       In vitro assays showing the antioxidant activity of LQFM289 would supplement the results presented in the manuscript.

Author Response

Reviewer #1

The studies described in the research article " LQFM289: Electrochemical and Computational Studies of a New Trimetozine Analogue for Anxiety Treatment" by Pereira et al have used electrochemical and Density Functional Theory (DFT) calculation approaches to test the potential of a novel analogue of trimetozine (TMZ) as a viable treatment option for anxiety disorders.

The following are some suggestions to improve the article:

  1. Line 57- Not all anxiolytic drugs target serotonin or norepinephrine. For example, Benzodiazepines facilitate GABA binding at GABA receptors.

Response: Thank you for this note. Your comment was reviewed.

  1. Line 64- Please change to “disorders”

Response: Thank you for this note. Your comment was reviewed.

  1. Line 185- Please correct “por”

Response: Thank you for this note. Your comment was reviewed.

  1. In vitro assays showing the antioxidant activity of LQFM289 would supplement the results presented in the manuscript.

Response: Thank you for this note. When there are high levels of ROEs, it can compromise the production of dopamine, norepinephrine, serotonin and nitric oxide, through tetrahydrobiopterin oxidation. Tetrahydrobiopterin is an importante coezime, which participates of biosynthesis of many neurotransmissor. So, compounds with antioxidant profile should help to reverse mood disorders.

  1. Neurauter, K. Schröcksnadel, S. Scholl-Bürgi, B. Sperner-Unterweger, C. Schubert, M. Ledochowski, D. Fuchs. Chronic immune stimulation correlates with reduced phenylalanine turnover. Curr Drug Metab. 2008, 9(7):622-7.

We will continue assessing the antioxidant profile of LQFM289 at in vitro models to confirm this hypothesis.

Reviewer 2 Report

In the current study the authors used an electrochemical and Density Functional Theory calculation approaches to investigate the potential of a novel analogue of trimetozine as a viable treatment option for anxiety disorders.

Some comments:

1. Please add what brings new your study considering that the anxiolytic activity of LQFM289 has already been studied (lines 77-83). Since the studies have reached the in vivo level, numerous in vitro studies have been performed.

Your reference No.[27] does not correspond to “In a previous work LQFM289 10 mg/kg induced anxiolytic-like activity in mice with-77 out eliciting motor incoordination” (lines 77-78). Please check.  

2. Please improve the quality of Figure 1B.  

3. pg 8, Give please details concerning the conection between your proposed electrooxidation mechanism and the anxiolytic activity.  

4. pg 10, lines 249-50: Add please the way in which the compounds Trimetozine and LQFM289 were obtained.

Minor editing of English language is required

Author Response

Reviewer #2

In the current study the authors used an electrochemical and Density Functional Theory calculation approaches to investigate the potential of a novel analogue of trimetozine as a viable treatment option for anxiety disorders.

Some comments:                    

  1. Please add what brings new your study considering that the anxiolytic activity of LQFM289 has already been studied (lines 77-83). Since the studies have reached the in vivo level, numerous in vitro studies have been performed.

Your reference No.[27] does not correspond to “In a previous work LQFM289 10 mg/kg induced anxiolytic-like activity in mice with-77 out eliciting motor incoordination” (lines 77-78). Please check. 

Response: Thank you for this note. The data suggesting LQFM289 is a multimodal compound. It showed anxiolity profile without interfere at with the animals' motor activity. Additionally it was able to reduce TNF-α, which is an anti-inflammatory cytokine. At a recente paper was possible to show the relationship between anxiety, anti-inflammatory and antioxidant compound.

Moreira, L.K.S. et al. LQFM212, a piperazine derivative, exhibits potential antioxidant effect as well as ameliorates LPS-induced behavioral, inflammatory and oxidative changes. Life Sci. 2023 Jan 1;312:121199. doi: 10.1016/j.lfs.2022.121199.

  1. Please improve the quality of Figure 1B.

Response: Thank you for this note. Figure 1B was reviewed.

  1. pg 8, Give please details concerning the conection between your proposed electrooxidation mechanism and the anxiolytic activity.

Response: Thank you for this note. At a recent paper we showed the relationship between anxiety, anti-inflammatory an antioxidant compound. At the moment, we are continuing the assessment of LQFM289 at anxiety, anti-inflammatory and antioxidant models in vivo. 

Moreira, L.K.S. et al. LQFM212, a piperazine derivative, exhibits potential antioxidant effect as well as ameliorates LPS-induced behavioral, inflammatory and oxidative changes. Life Sci. 2023 Jan 1;312:121199. doi: 10.1016/j.lfs.2022.121199.

The proposed electro-oxidation mechanism is related to observed voltametric data at experiments and based at literature [48].

  1. pg 10, lines 249-50: Add please the way in which the compounds Trimetozine and LQFM289 were obtained.

Response: Thanks for the note. The compounds were obtained as described at refernce [28].

Reviewer 3 Report

1. Please change "According to the World Health Organization" to "According to the World Health Organization (WHO)".

2. The introduction does not explain the broader context of the research. While it mentions anxiety disorders and oxidative stress, it doesn't provide a clear rationale for why investigating a novel analogue of trimetozine (TMZ) is relevant or important compared to current anxiolytic medications.

3. Please separate the "Results" and "Discussion" sections rather than combine them.

4. "... the tree lowest" - spelling error.

5. The nature of replication in the experimental design is rather unclear, and the assessment of uncertainty in the reported measurement is absent or unclear at parts.

6. The study describes the findings (two anodic reactions) but doesn't go far enough to elucidate their significance. Why are these two steps important? How do they relate to the compound's potential as a therapeutic agent?

7. The mention of computational studies agreeing with electrochemical findings is a positive note, but it does not delve into the significance of this agreement. Are there studies to support this? How do these findings enhance our overall understanding of the compound's behavior or its potential as a treatment?

8. There is no mention of any limitations or potential shortcomings of the study.

9. Suggest including a sentence or two about potential next steps in the research in the Discussion section. This could involve further studies, refinement of the compound, or exploring its therapeutic effects in preclinical or clinical settings.

10. Suggest making the conclusion a single concise paragraph.

Moderate edits needed.

Author Response

Reviewer #3             

  1. Please change "According to the World Health Organization" to "According to the World Health Organization (WHO)".

Response: Thank you for this note. Your comment was reviewed.

  1. The introduction does not explain the broader context of the research. While it mentions anxiety disorders and oxidative stress, it doesn't provide a clear rationale for why investigating a novel analogue of trimetozine (TMZ) is relevant or important compared to current anxiolytic medications.

Response: Thank you for this note. When there are high levels of ROEs, it can compromise the production of dopamine, norepinephrine, serotonin and nitric oxide, through tetrahydrobiopterin oxidation. Tetrahydrobiopterin is an importante coezime, which participates of biosynthesis of many neurotransmissor. So, compounds with antioxidant profile should help to reverse mood disorders.

  1. Neurauter, K. Schröcksnadel, S. Scholl-Bürgi, B. Sperner-Unterweger, C. Schubert, M. Ledochowski, D. Fuchs. Chronic immune stimulation correlates with reduced phenylalanine turnover. Curr Drug Metab. 2008, 9(7):622-7.

We will continue assessing the antioxidant profile of LQFM289 at in vitro models to confirm this hypothesis.

At a recente paper we showed the relationship between anxiety, anti-inflammatory an antioxidant compound. At the moment, we are continuing the assessment of LQFM289 at anxiety, anti-inflammatory and antioxidant models in vivo. 

Moreira, L.K.S. et al. LQFM212, a piperazine derivative, exhibits potential antioxidant effect as well as ameliorates LPS-induced behavioral, inflammatory and oxidative changes. Life Sci. 2023 Jan 1;312:121199. doi: 10.1016/j.lfs.2022.121199.

  1. Please separate the "Results" and "Discussion" sections rather than combine them.

Response: The requeriment was attended.

  1. "... the tree lowest" - spelling error.

Response: Thank you for this note. Your comment was reviewed.

  1. The nature of replication in the experimental design is rather unclear, and the assessment of uncertainty in the reported measurement is absent or unclear at parts.

Response: Thank you for this note. Your comment was reviewed.

  1. The study describes the findings (two anodic reactions) but doesn't go far enough to elucidate their significance. Why are these two steps important? How do they relate to the compound's potential as a therapeutic agent?

Response: Thank you for this note. We did some comments at question 2.

Also some lost of reproductibility can be attributes to chaotic electropolierization reactions (Figures 2A, 7 and 8).

  1. The mention of computational studies agreeing with electrochemical findings is a positive note, but it does not delve into the significance of this agreement. Are there studies to support this? How do these findings enhance our overall understanding of the compound's behavior or its potential as a treatment?

Response: Thank you for this note. The Docking of LQFM289 at benzodiazepine binding sites was conducted at reference [28]. In this paper the computational studies showed relationship with electrochemical behaviour.

Indeed, since our first paper in this line “E. S. Gil, C. H. Andrade, N. L. Barbosa, R. C. Braga, S. H. P. Serrano, Cyclic voltammetry and computational chemistry studies on the evaluation of the redox behavior of parabens and other analogues, J. Braz, Chem. Soc. 23 (2012) 565.” there are many other papers that have been exploring computational studies with voltammetruc data.

https://hrcak.srce.hr/clanak/291924, https://doi.org/10.1021/acs.jpcc.0c07591, https://doi.org/10.1016/j.molliq.2018.11.147, https://doi.org/10.1039/B803717E, https://doi.org/10.1016/j.molstruc.2022.134662, https://doi.org/10.1016/j.jelechem.2014.12.024, https://doi.org/10.1021/jp400021u, https://doi.org/10.3390/ph12030116, https://doi.org/10.1016/j.electacta.2018.02.128, http://dx.doi.org/10.2139/ssrn.4385067, ….

  1. There is no mention of any limitations or potential shortcomings of the study.

Response: Thank you for this note. Your comment was reviewed.

  1. Suggest including a sentence or two about potential next steps in the research in the Discussion section. This could involve further studies, refinement of the compound, or exploring its therapeutic effects in preclinical or clinical settings.

Response: Thank you for this note. Your comment was reviewed.

The computational data and electrochemical characterization are useful to indicate the molecular moieties most vulnerable to undergo redox reactions.

  1. Suggest making the conclusion a single concise paragraph.

Response: Thank you for this note. Your comment was reviewed.

Reviewer 4 Report

Current report investigated LQFM289 derived from molecular hybridization (BHT-TMZ) as synthetic antioxidant for psychological disorders. I like to give the following comments.

1.      Association of LQFM289 with growth differentiation factor 11 (GDF11) was not introduced. Why?

2.      The anxiolytic drugs with antioxidant-like properties need to introduce in clear.

3.      Docking of LQFM289 with benzodiazepine binding sites was not conducted. Why?

4.      In Figure 6, molecules superposition between (b) and (c) needs to explain in the legends.

5.      Computational studies need the supports from biological data.

6.      The voltammetric methods shown to be a useful tool for redox stability and antioxidant activity in conclusion. However, it was not conducted in results.

7.      Limitation(s) of current report will be useful.

It seems better to check through professional editing.

Author Response

Reviewer #4

Current report investigated LQFM289 derived from molecular hybridization (BHT-TMZ) as synthetic antioxidant for psychological disorders. I like to give the following comments.                

  1. Association of LQFM289 with growth differentiation factor 11 (GDF11) was not introduced. Why?

Response: Thank you for this note. The GDF11 is an anti-inflammatory cytokine, which is in study to treatment of CNS disease.

  1. The anxiolytic drugs with antioxidant-like properties need to introduce in clear.

Response: Thank you for this note. Your comment was reviewed.

  1. Docking of LQFM289 with benzodiazepine binding sites was not conducted. Why?

Response: Thank you for this note. The Docking of LQFM289 at benzodiazepine binding sites was conducted at reference [28].

  1. In Figure 6, molecules superposition between (b) and (c) needs to explain in the legends.

Response: Thanks for the note. Your comment was reviewed.

  1. Computational studies need the supports from biological data.

Response: Thanks for the note. The Docking of LQFM289 at benzodiazepine binding sites was conducted at reference [28]. In this paper the computational studies showed relationship with electrochemical behaviour.

  1. The voltammetric methods shown to be a useful tool for redox stability and antioxidant activity in conclusion. However, it was not conducted in results.

Response: Thank you for this note. Your comment was reviewed.

  1. Limitation(s) of current report will be useful.

Response: Thank you for this note. Your comment was reviewed.

Round 2

Reviewer 3 Report

Several of the comments were supposedly 'noted' by the authors but not acted upon at all.

1. Introduction needs improvement for paragraphing and writing.

2. The nature of replication in the experimental design is rather unclear, and the assessment of uncertainty in the reported measurement is absent or unclear at parts.

3. There is no mention of any limitations or potential shortcomings of the study.

4. How do you translate the findings clinically? What are the next steps?

5. Suggest making the conclusion a single concise paragraph.

Moderate edits needed.

Author Response

Several of the comments were supposedly 'noted' by the authors but not acted upon at all.

  1. Introduction needs improvement for paragraphing and writing.

Answer: Introduction was improved, the missing comment on DFT in the introductory section was added

  1. The nature of replication in the experimental design is rather unclear, and the assessment of uncertainty in the reported measurement is absent or unclear at parts.

Response: Thank you for this note. Your comment was reviewed.

  1. There is no mention of any limitations or potential shortcomings of the study.

Answer: A brief mention of the limitations of the employed methods was written in the conclusion section.

  1. How do you translate the findings clinically? What are the next steps?

Answer: Nowadays, we are continuing the assessment pharmacological of LQFM289 through anxiety, neuroinflammation and oxidative stress models. In turn, we also will go to assess the toxicological profile of LQFM289. This information will be very important to conduce the design of new LQFM289’s analogues.

  1. Suggest making the conclusion a single concise paragraph.

Answer: The conclusion was rearranged to be contained in a more concise and better explained paragraph.

Round 3

Reviewer 3 Report

'Departament' is spelled wrongly.

Moderate edits still.